# Inhibitory interactions promote frequent bistability among competing bacteria

Erik S. Wright[1] & Kalin H. Vetsigian[1]

It is largely unknown how the process of microbial community assembly is affected by the order of species arrival, initial species abundances and interactions between species. A minimal way of capturing competitive abilities in a frequency-dependent manner is with an invasibility network specifying whether a species at low abundance can increase in frequency in an environment dominated by another species. Here, using a panel of prolific small-molecule producers and a habitat with feast-and-famine cycles, we show that the most abundant strain can often exclude other strains—resulting in bistability between pairs of strains. Instead of a single winner, the empirically determined invasibility network is ruled by multiple strains that cannot invade each other, and does not contain loops of cyclic dominance. Antibiotic inhibition contributes to bistability by helping producers resist invasions while at high abundance and by reducing producers' ability to invade when at low abundance.

[1] Department of Bacteriology, Wisconsin Institute for Discovery, University of Wisconsin-Madison, 330 N. Orchard Street, Madison, Wisconsin 53715, USA. Correspondence and requests for materials should be addressed to K.H.V. (email: kalin@discovery.wisc.edu).

Microbes undergo high rates of dispersal and intermixing in nature but are constrained in their ability to colonize new habitats by environmental filtering and local competition[1]. After undergoing the process of community assembly, established microbial communities are continually challenged by immigrants from other microbial communities that have the potential to upset the existing community balance. An understanding of the assembly and resilience of communities therefore requires knowledge of interactions among established community residents and outsiders that may arrive at low abundance through dispersal[2,3]. Invasion experiments are a minimal way of capturing such competitive outcomes at the two extremes of relative abundance (Fig. 1). These pairwise relationships can be assembled into an invasibility network, which characterizes the effective interactions between microbes and their frequency dependence[4].

Even a basic knowledge of the statistical properties of invasibility networks for microbes from similar habitats can greatly enhance our understanding of the processes that structure natural communities[5]. The simplest expectation, based on the competitive exclusion principle[6], is that most microbial strains can be ordered according to their competitive ability, which would result in a hierarchical network composed of asymmetrical invasions (Fig. 1) and dynamics dominated by the survival of the fittest for any particular environment[7]. On the other hand, networks exhibiting cyclic dominance (as in rock-paper-scissor games) can lead to diversity maintenance or alternating winners[8–10]. Widespread negative frequency-dependent selection would imply an advantage for rare variants, which can promote diversity, whereas positive frequency-dependent selection can lead to alternate stable states[11] and historical contingency[12,13]. Despite their utility, the statistical properties of invasibility networks are generally unknown[14–16].

To start filling this knowledge gap, we determined the invasion and inhibition networks for a diverse panel of bacteria from the genus *Streptomyces* (Supplementary Fig. 1), most of which were isolated from neighbouring grains in the same soil sample (see the Methods for details). These bacteria are ubiquitous in soil and are prolific producers of antibiotics and other secondary metabolites when grown on a solid substrate[8]. Understanding the ecological consequences of diverse secondary metabolites has been a major challenge[17,18], inspiring many theoretical and experimental works[10,19–21]. However, the role of secreted bioactive compounds in generating intransitive or frequency-dependent relationships has not been investigated systematically among large collections of microbes, particularly in unmixed environments, in which secreted molecules stay close to their producers.

We find that 'survival of the common' is nearly as widespread as 'survival of the fittest' among competing soil bacteria from the genus *Streptomyces*. The winner of a pairwise competition is often the species that starts at high initial abundance, making it impossible to completely rank the species based on their competitive ability. Instead of a single winner, the tournament between bacteria results in multiple winners that are in bistable relationships with each other. We also find that inhibitory interactions between species are an important factor shaping the network of invasions, and such inhibitory interactions promote 'survival of the common'. These findings have several immediate implications for how we view the assembly, structuring and diversity of microbial communities. They indicate that pairwise interactions lead to inherent nonlinearities that predispose communities towards multiple stable states. This may make microbial communities intrinsically sensitive to initial conditions during community assembly but, at the same time, could make them more resistant to change once they are established. Survival of the common may also promote mosaic spatial distributions

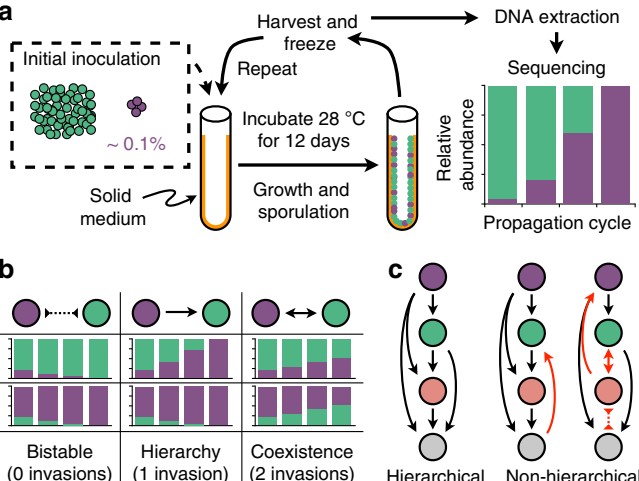

**Figure 1 | Scheme for measuring pairwise invasions and potential outcomes.** (**a**) Pairs of bacterial strains were added to test tubes at vastly different initial abundances and propagated for three cycles. Their relative abundance was quantified with high-throughput sequencing. (**b**) Each strain from a pair was competed twice, as either resident (high abundance) or invader (low abundance). The three potential outcomes are: bistability if neither resident was invaded, hierarchy if one resident was invaded and coexistence if both residents were invaded. (**c**) The invasion network for a panel of strains may be either completely hierarchical, where strains can be ranked by relative fitness in a way that explains all pairwise outcomes, partly hierarchical with a small fraction of non-hierarchical features, or essentially non-hierarchical.

with different populations dominating different patches or microbial hosts despite similar abiotic conditions.

## Results

**Frequent bistable relationships between pairs of strains.** To measure invasion, we inoculated a pair of strains at vastly different initial abundances inside a thin layer of solid (agar) defined medium and allowed them to grow and sporulate (Fig. 1a). Offspring spores were then collected from the surface of the agar and then used to inoculate another propagation cycle or determine relative abundances with high-throughput sequencing (see the Methods for details). After three propagation cycles, strains were said to invade if they had increased in abundance to at least 1% of the total community. Typically, invasions occurred rapidly, and the invader had almost completely displaced the resident within one or two propagation cycles (Supplementary Fig. 2).

We began by analysing pairwise features of the invasion matrix. Invasions were highly repeatable, as we only observed a single difference between 32 replicate competitions performed with strain #1 (Fig. 2a). Overall, 31% of pairwise competitions resulted in an invasion (Fig. 2b). No strain was invaded by all other strains in the panel, although one strain (#14) was invaded by all but two others. Three strains were not invaded by any other strain, indicating that the strains cannot be ordered in a strict hierarchy. Six of seven cases of mutual invasion included strain #1 (Fig. 2b), which was also the most distantly related strain as it belongs to a separate genus (Fig. 2a). Mutual invasions are expected to lead to coexistence because neither strain can reach a low enough abundance that it is unable to recover. Accordingly, in all seven cases, the pairs of mutually invading strains were both found to be present at the end of three propagation cycles.

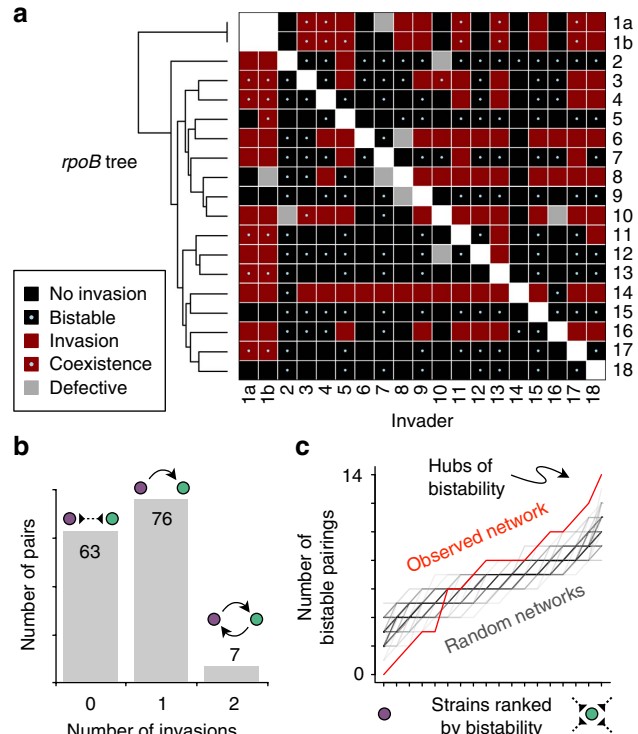

**Figure 2 | Widespread bistability in pairwise invasions.** (**a**) Pairwise invasion matrix for a panel of 18 diverse *Streptomyces* strains. Strains are sorted by phylogeny constructed from partial *rpoB* gene sequences. Strain #1 is present in two replicas (labeled 1a,b). (**b**) Bistable pairings, in which two strains cannot invade each other, were a dominant feature of the invasion matrix. Coexistence was less frequent and mostly limited to strain #1, which was also the most phylogenetically distinct strain. (**c**) A few strains were involved in many bistable pairings. These 'hubs of bistability' were more frequent than in randomized matrices with the same number of each type of pairwise link ($P = 1.7e - 4$).

In sharp contrast to the low number of mutual invasions, there were 63 mutually non-invading pairs of strains, where the most abundant strain was able to hold its ground against the less abundant (Fig. 2b). These bistable links centred on a small subset of strains that rarely invaded others and were rarely invaded by others, and therefore acted as 'hubs of bistability' (Fig. 2c).

**Partly hierarchical invasion network with multiple winners.** We next characterized triplet motifs in the invasion network relative to random networks with the same number of each type of pairwise link. We observed a strong enrichment for transitivity of hierarchy: given that strain A invades B and B invades C, A most likely also invades C (Fig. 3a). Surprisingly, we did not observe a single instance of the 'rock-paper-scissors' dynamic (C invading A). Similar to hierarchy, bistable links were also greatly enriched for transitivity (Supplementary Fig. 3).

The strong enrichment for transitivity of hierarchy motivated us to order species according to their competitive ability by determining the rank assignments that are most congruent with the observed invasibility network. To accomplish this, we developed a simple scoring scheme that rewarded invasions directed down the hierarchy, and penalized invasions going against the hierarchy (Fig. 3b). Under this scoring scheme, the optimal hierarchy placed the strains into seven levels (Fig. 3c). Only a few links were directed against the hierarchy, and most of them were due to mutual invasions with strain #1. Instead of a

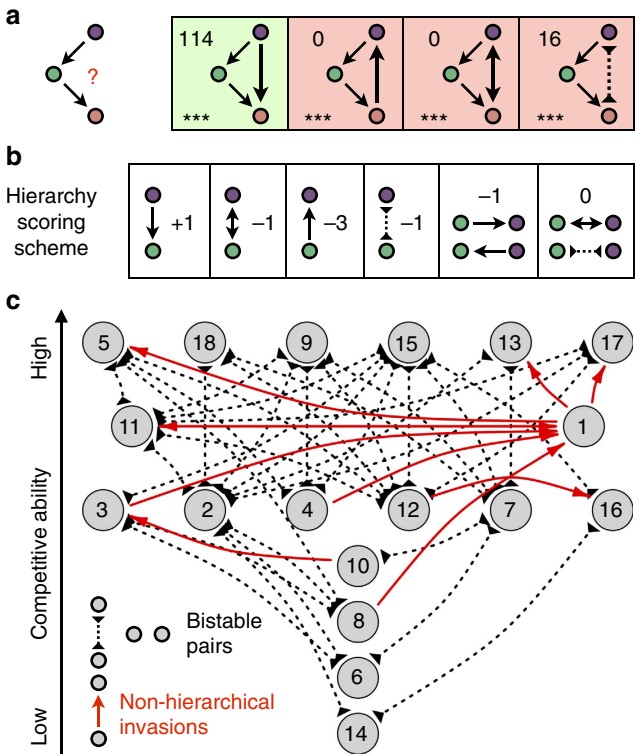

**Figure 3 | The invasion network is partly hierarchical with multiple strains in the top level exhibiting bistable relationships with each other.** (**a**) Enrichment (green) or depletion (pink) of different triplet motifs relative to randomized networks preserving the number of each pairwise motif (Supplementary Fig. 3). Number of occurrences for each motif are given in the upper-left corner. Transitive invasions (leftmost motif) were highly enriched (***,$P < 1e - 6$), whereas the three intransitive motifs were highly depleted (***,$P < 1e - 6$). (**b**) The scoring scheme used in assigning hierarchy levels to strains rewards invasions pointing down the hierarchy and penalizes invasions directed against the hierarchy. (**c**) The invasion network overlaid on the hierarchy assignments that maximize the score in **b**. Strains were placed into seven hierarchy levels with six strains at the top level exhibiting bistable relationships with each other. Invasions going down the hierarchy are not shown, while others are shown in red. Bistability is denoted by a missing link between strains at the same level or by a dashed line for strains at different levels.

single fittest strain, six strains were tied for the top ranking, and, remarkably, all of them were in bistable relationships with each other. Thus, the hierarchical structure of the invasibility network revealed six mutually exclusive winners. Fascinatingly, there was considerable bistability between strains belonging to different hierarchical levels. In many of these cases, an invader strain from the top of the hierarchy could form visible colonies or inhibition zones against a strain below it in the hierarchy, yet ultimately failed to invade (Supplementary Fig. 4). Furthermore, although strains exhibited widely different yields and growth rates in our experimental system, both measures were uncorrelated ($R^2 = 0.006$ and $0.174$, respectively) with hierarchy level (Supplementary Fig. 5).

**Inhibitory interactions promote bistability.** Given that bacteria from the genus *Streptomyces* are prolific antibiotic and siderophore[22–24] producers, we hypothesized that inhibitory interactions played a major role in determining the hierarchy

and generating bistability. To systematically examine the role of inhibition, we measured each strain's ability to prevent the sporulation of other strains (Supplementary Fig. 6). The data revealed a strong tendency for inhibitions to point down the hierarchy (Fig. 4a), which is consistent with the notion that inhibition provides a competitive advantage[17]. We found that strains were extremely unlikely to be invaded by strains they inhibit ($P = 1e - 6$, Fig. 4b), and there was a small increase in the probability to invade if inhibition was present ($P = 0.02$, Supplementary Fig. 7a).

An alternative explanation for why inhibitions point down the hierarchy is that the hierarchy and the direction of inhibition are shaped by a common factor. For example, species better adapted to the growth medium may tend to outcompete other species while also having a head start in antibiotic production. To investigate this possibility, we reconstructed the hierarchy using only species pairs without inhibition. The new hierarchy was similar to the original one ($R^2 = 0.916$; Supplementary Table 1) and still exhibited a pronounced tendency for downward inhibitions (Supplementary Fig. 8), suggesting that a common factor was at least partially responsible for both the hierarchy and the direction of inhibition.

To control for this confounding effect, we recomputed the correlations between inhibition and invasion while focusing on pairs of strains with similar differences in hierarchy levels (see the Methods for details). Unexpectedly, this revealed that strains that inhibit other strains are significantly less likely to invade (Fig. 4c, $P = 1.3e - 4$). Hence, the small apparent increase in probability to invade as an inhibitor was entirely due to the tendency of inhibition to go down the hierarchy. This finding is consistent with the notion that investment in public goods might be counterproductive when the producers are at low abundance.

We concluded that although inhibitions likely help strains resist invasions at high abundance, they also reduce their probability to invade when at low abundance. The combination of these effects leads to a higher probability of bistability in pairs of strains in which there is inhibition ($P = 0.028$). This enrichment was particularly pronounced after removing the two outlier strains at the bottom of the hierarchy, which are inhibited and outcompeted by almost everyone (Fig. 4d, $P = 0.002$). This indicates that inhibition is one of the mechanisms promoting bistability.

## Discussion

The finding of frequent bistability among *Streptomyces* isolated from the same environment is consistent with a recent theoretical work demonstrating that the counteraction of antibiotic production and degradation can lead to stable coexistence of many bacteria with different production and degradation capabilities[10]. Extending this previous work, we proved that multi-species communities coexisting through this mechanism must contain bistable pairs (see the Methods for details). Thus, frequent pairwise bistability is expected theoretically among sets of three or more strains coexisting through this mechanism. Although it is unknown whether the strains used in this study would coexist in larger communities, this study demonstrates that, in addition to being prolific producers and degraders of antibiotics, *Streptomyces* exhibit the frequency-dependent relationships necessary for coexistence through an interplay between antibiotic production and degradation.

Although we found a link between inhibition and bistability, it is important to note that many bistable pairs had no measurable inhibition. These may be cases where inhibition was below our detection limit or was only partial and therefore did not lead to a visible zone of clearing. Bistability in these cases may also be due

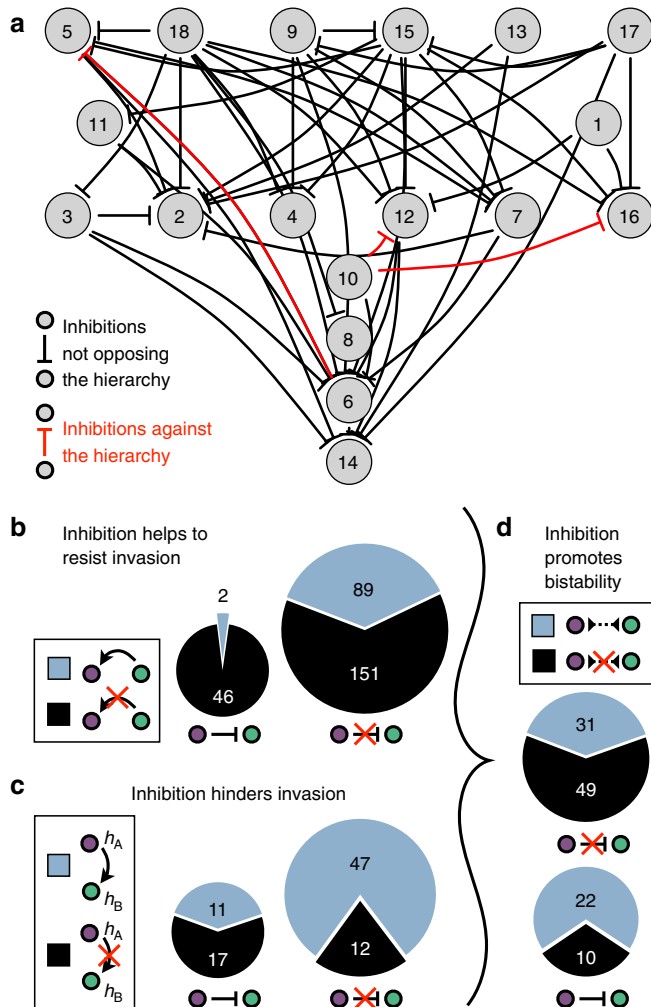

**Figure 4 | Antibiotic inhibition helps bistability.** (**a**) Inhibitions in the cross-streaking assay (Supplementary Fig. 6) overlaid on the invasion hierarchy. Almost all inhibitions were directed down the hierarchy (black inhibition arrows) and only a few were directed against the hierarchy (red inhibition arrows). (**b**) The number of pairs with invasions (blue) and non-invasions (black) are shown for cases in which the invader is inhibited (left pie-chart) or not inhibited (right pie-chart). The vastly different fraction of invasions in the two pie-charts indicates that inhibition greatly assists residents in resisting invasion. (**c**) The number of pairs with invasions (blue) and non-invasions (black) are shown for cases in which the resident is inhibited (left pie-chart) or not inhibited (right pie-chart). Only pairs in which the invader is at a higher hierarchy level ($3 \geq h_A - h_B > 0$) are considered to control for the tendency of inhibitions to point down the hierarchy. (**d**) The number of bistable (blue) and non-bistable (black) pairs is shown for pairs with inhibition (bottom pie-chart) and without inhibition (top pie-chart). Strains #14 and #6 at the bottom of the hierarchy were not included, as they were invaded by almost all other strains. A significant enrichment for bistability is evident among pairs with an inhibitory interaction.

to other phenomena of intra-species cooperation that incur a cost of rarity[25], including quorum sensing[26] and the secretion of public goods such as extracellular enzymes. The existence of multiple life-stages, such as the germination—mycellium growth—sporulation lifecycle of *Streptomyces*, can also facilitate bistability[27]. Delineating the relative contribution of each of these potential causes of bistability in different microbial systems offers an interesting topic for future research.

Frequent bistability between bacterial strains has major implications for community assembly and structuring. First, it implies that the order of species arrival to a new environment could strongly influence the long-term community composition. In particular, pairwise bistability is expected to propagate to multistability in communities with more than two species. Furthermore, bistability may lead to an extreme sensitivity to initial abundances when many species arrive simultaneously, while also making established communities more resistant to invasions. These effects may manifest spatially through the generation of mosaic microbial distributions in which different communities are maintained in different spatial locations[28] or in different individuals from a plant or animal host species. In particular, frequent bistability may help explain the recent finding that soil bacterial and fungal communities exhibit higher levels of dissimilarity across locations than expected from models assuming that environmental filtering and dispersal are the primary drivers of community assembly[29,30]. Finally, on longer timescales, positive frequency-dependent selection could encourage the emergence of discrete microbial types as is typical for larger organisms[31], rather than a continuous spectrum of forms.

We expect the finding of widespread survival of the common to generalize to other microorganisms that produce strain-specific public goods, particularly when they are grown in unmixed environments and have strong resource overlaps[19,32]. Further insights into the survival of the common and its ramifications for ecosystems are likely to emerge from continued research on multi-species microcosm communities and gnotobiotic organisms.

## Methods

**Isolation of Streptomycetes.** The panel of 18 strains (Supplementary Table 1) used in this study included 13 *Streptomyces* isolates originating from the same soil sample. This sample was collected from the University of Wisconsin-West Madison Agricultural Research Station on 10 June 2014. The soil in the collection area was composed of Troxel silt loam at 1–3% slopes. A soil core was collected using a sterile 50 ml conical tube (VWR). A 0.25-g piece of soil was extracted from a depth of 4 cm below the surface. This soil was separated into individual grains ($\sim$1 mg), each of which was used to inoculate a Petri dish containing Actinomycete Isolation Agar (4 g l$^{-1}$ sodium propionate, 10 g l$^{-1}$ soluble starch, 0.4 g l$^{-1}$ sodium caseinate, 2 g l$^{-1}$ KNO$_3$, 2 g l$^{-1}$ NaCl, 2 g l$^{-1}$ K$_2$HPO$_4$, 0.05 g l$^{-1}$ MgSO$_4$, 0.02 g l$^{-1}$ CaCO$_3$, 0.01 g l$^{-1}$ FeSO$_4$, 18 g l$^{-1}$ Agar; pH adjusted to 7.5; to prevent fungal growth, cyclohexamide added after autoclaving to reach 50 mg l$^{-1}$).

Separated isolates on each Petri dish were automatically detected and pinned into individual wells of a 96-well plate containing Actinomycete Isolation Agar medium using an automatic colony picker (Hudson). To distinguish isolates, we sequenced a 935–base-pair region of their DNA-directed RNA polymerase subunit β (*rpoB*) gene, which is commonly used as a species-level phylogenetic marker for *Streptomyces*[33]. We identified 28 distinct *Streptomyces* isolates, and selected 13 for invasion experiments based on having sufficient growth for sustained propagation on the defined medium used in the invasion experiment. These 13 isolates were named by their soil grain and microplate well. For example, sp. *S25E2* and sp. *S25H8* were both isolated from the same soil grain (S25). The remaining five strains used in this study were collected from various sources (Supplementary Table 1). Using the R[34] package DECIPHER[35], these 18 strains' *rpoB* sequences were aligned to those from related species in order to create the phylogenetic tree shown in Supplementary Fig. 1.

**Invasion experiments.** Before the start of the invasion experiment, the 18 strains were individually propagated for three growth cycles to reach equilibrium concentration on the defined medium (see below) used in the invasion experiments. A 50-μl aliquot of the *resident* strain at its equilibrium concentration was added to 80 μl of the diluted *invader* in the initial inoculum. The concentration in colony-forming units (CFUs) of each spore stock was determined using standard dilution plating. Each strain was diluted to achieve a target concentration of 100 CFUs per tube as *invader* in the first inoculation. The *invader* strains were again counted after dilution to determine their actual concentration in the tubes, and were generally close to the desired cell count (Supplementary Table 1). The *invader* concentration was typically less than 0.1% of the *resident* cells at the beginning of the first growth cycle.

Communities were grown in 18 × 150 mm glass tubes (Fisher Scientific) containing 4.5 ml of defined medium (10 g starch, 0.4 g proline, 0.4 g asparagine, 2 g l$^{-1}$ KNO$_3$, 2 g l$^{-1}$ NaCl, 2 g l$^{-1}$ K$_2$HPO$_4$, 0.05 g l$^{-1}$ MgSO$_4$, 0.01 g l$^{-1}$ FeSO$_4$, 25 g l$^{-1}$ agar; adjust pH to 7.0 with 5 N NaOH). Inocula were injected into molten agar ($\sim$50 °C), vortexed briefly to mix, and rolled horizontally (along their long-axis) at 1,800 r.p.m. for 45 s under high air flow to coat the inside of the tube with a thin ($\sim$0.8 mm) layer of solid agar. The inside of the tube was therefore hollow, which allowed oxygen to reach the cells. Tubes were stored upright at 28 °C for 12 days before harvesting.

Tubes were harvested by first adding 4 ml of sterile 2 mm glass beads (Chemglass Life Sciences) and vortexing for 10 s to remove the hydrophobic spores from the agar surface. A 5-ml filter-sterilized solution of 0.1% Tween-80 and 20% glycerol was added to each tube and vortexed for 10 s. An aliquot of 1.7 ml of each community was collected and frozen at −80 °C. Communities were frozen between growth cycles to ensure that they could be grown from a consistent state in future uses. In subsequent growths, 100 μl of each previously harvested community was used to inoculate its next propagation cycle (1/50th dilution per cycle). This process was repeated until the communities had been grown for three propagation cycles.

**DNA extraction and sequencing.** To efficiently extract DNA from harvested biomass, it was necessary to first germinate the spores. A 100-μl aliquot of each community was grown in a test tube with 2 ml of the defined medium without agar. The liquid cultures were incubated while shaking for 40 h at 28 °C. Next, the cultures were centrifuged at 1,000 relative centrifugal force (r.c.f) for 10 min to pellet the cells. A 1.7-ml volume of supernatant was removed, the remaining volume was vortexed and 200 μl of the concentrated mycelium was transferred to a 0.2-ml thin-wall tube (Corning). These tubes were sonicated at 100% amplitude for 60 s using a Model 505 Sonicator with Cup Horn (QSonica). After sonication, the samples were centrifuged, and the supernatant containing DNA was used as template for PCR amplification.

Primers were designed to optimally differentiate an 80 nucleotide region of the *rpoB* sequence of all 18 strains using DesignSignatures[36] (Supplementary Table 2). The targeted *rpoB* region differed by a minimum of 4 nucleotides between all strain pairs (median of 12 nucleotides different). Extracted DNA was first amplified with primers targeting sites that were universal to all species, diluted and then re-amplified with barcoded primers (Supplementary Table 2). This two-step process can decrease amplification bias during PCR[37] and mitigate the amplification of PCR artefacts associated with long adapter primers. The two PCR reactions consisted of a 2-min denaturation step at 95 °C, followed by 45 and 25 cycles, respectively, of 20 s at 98 °C, 15 s at 67 °C and 15 s at 80 °C. Each PCR reaction was followed by a melt curve from 60 to 95 °C in 0.5 °C increments every 10 s to confirm the expected melt peak. The PCR reaction contained 2.5 μl of iQ Supermix (Bio-Rad), 0.4 μl of 5 μM forward and reverse primer, 0.5 μl of DNA template and 1.6 μl reagent grade H$_2$O per sample. The 177 base pair product of the first PCR reaction was diluted 1,000-fold for use as template in the second reaction with barcoded primers. Barcoded primers were staggered by inserting 0–3 additional nucleotides before the sequencing read to help with randomization for phasing[38]. Groups of 24 PCR products (2.5 μl per sample) were pooled into 50 μl of 10 × TBE buffer on ice. This mix of PCR products was separated by length in a 1% agarose gel. The band matching the desired length ($\sim$310 nucleotides) was excised from the gel, and purified with the Wizard SV-Gel and PCR Cleanup System (Promega). All samples were sequenced by the UW-Madison Biotechnology Center with an Illumina Hi-Seq in rapid mode.

**Determination of presence or absence of the invader.** Barcoded primers contained Illumina adapters with i5 and i7 index sequences that were unique to each community. These index sequences allowed for de-multiplexing of the samples by exactly matching the pair of eight nucleotide index sequences to the unique combination belonging to each community. By using 25 different i5 and i7 index sequences, we were able to multiplex up to 625 samples in the same sequencing lane. This approach typically resulted in more than 10,000 reads of 101 nucleotides per community. The reads were exactly matched to the known *rpoB* sequences for the panel of 18 strains in order to count the relative abundance of each strain. Analysis of read counts was performed with the R[34] package Biostrings[39].

Based on the known species that could be present in each community, we identified a cross-indexing error during sequencing in which a wrong i7 index was associated with a read at a rate of $\sim$0.1%. The error rate varied between i7 indices in proportion to the total number of reads with a given i7 index. This simple statistical model explained well the unexpected reads and enabled background subtraction. The background level of reads for each species and community was determined by summing the contributions from other index pairs sharing the same i5 index, and resulted in effective background levels of less than 1% per species in a community. This background model was further confirmed by the distribution of reads belonging to a strain that was not present in any community and had been amplified separately with a unique i5 index and i7 index pair using primers that were independently synthesized.

Strains were said to invade if they had increased in frequency and reached at least 1% of the total community after subtracting the background. Strains not

appearing above the background level were considered below the detection limit and marked as non-invasions. In two of the communities, the invaders appeared only slightly above the background read level and were marked as defective and not considered in the analyses. Other cases of defective communities were due to experimental failure.

We further assessed a subset of eight bistable strain pairs using quantitative PCR (Supplementary Table 3), which has a larger dynamic range than high-throughput sequencing. Primers were designed that were specific to each strain using the R[34] package DECIPHER[40] (Supplementary Table 2). The resident and invader were targeted for amplification in two separate PCR reactions for each community, corresponding to 32 different reactions in total. The reaction conditions were identical to those described above, except that we used tenfold larger reactions (50 μl total). In 45 PCR cycles, we observed early amplification of all 16 residents (Supplementary Table 3), followed by delayed amplification with primers targeting the invader. Using melt curves, gel runs and Sanger sequencing, we were able to confirm that all invader amplifications were standard PCR artefacts attributable to the high number of PCR cycles and large reaction size. Therefore, in all eight bistable pairs, we were unable to detect the presence of either invader after three growth cycles.

**Inhibition experiments.** To quantify inhibition, we used a standard cross-streaking test on a Petri dish (Supplementary Fig. 6). First, all strains were grown at high density and allowed to sporulate on defined medium (see above). We used a flat rectangular ($8 \times 81$ mm) aluminum pinning tool to transfer spores from a *majority* strain to the centre of a new plate with a thin layer of defined medium (5 ml per 88 mm diameter Petri dish). A cross-streak of a *minority* strain was plated perpendicular to the *majority* strain using a 60-mm microscope coverslip with a thickness of ~0.15 mm (VWR). Five *minority* strains, separated by 1 cm, were plated in parallel across the same *majority* strain on a single plate. The strains were allowed to grow for 12 days at 28 °C, and imaged periodically with a flatbed scanner. We used an in-house R[34] script to quantify the distance between the *majority* strain and where the *minority* strain had sporulated in the obtained images. The *majority* strain was defined as inhibiting the *minority* strain if it prevented sporulation within a distance of 1 mm or greater.

**Measurement of growth rate and yield.** To measure growth rate, strains were grown on a thin layer of defined medium (see above) for 43 h and imaged under a microscope. The surface area of three to seven separated colonies was determined with ImageJ[41], averaged and scaled to units of mm² for plotting in Supplementary Fig. 5. To measure yield, each strain was grown alone for three sequential growth cycles as in the invasion experiments described above. The final concentration in CFU per μl at the end of all three growth cycles was determined by standard tenfold dilution plating and counting.

**Hubs of bistability.** Bistability occurs when strain $i$ does not invade $j$ and $j$ does not invade $i$, corresponding to non-invasions on opposite sides of the diagonal in the invasion matrix. The strains can be sorted by their number of bistable interactions, ranging from 0 (strain #1) to 14 (strain #2). We compared the number of bistable pairings per strain to that of a random invasion network with the same total number of each type of pairwise link. To determine the statistical significance of the observed hubs, we calculated the fraction of random networks for which the sum of bistabilities associated with the top three strains was greater than or equal to that of the three most bistable strains in the measured invasion network.

**Analysis of triplet motifs in the invasion network.** There are 16 distinct triplet motifs possible when allowing for bistability (0 invasions), hierarchy (1 invasion) and coexistence (2 invasions). To assess the statistical significance of the different triplet motifs in the invasion network, we compared their frequency with those expected for random networks with the same number of pairwise links (0, 1 or 2 invasions; Supplementary Fig. 3).

**Determination of the invasion network hierarchy.** A simple scoring model was constructed to assess a given ranking of strains based on the invasion matrix, as described in the Results. This model was based on a previous study of assigning hierarchy to directed networks[42]. The optimal ranking of strains was determined using the R[34] package rgenoud[43] for integer genetic optimization. We repeated the optimization procedure 1,000 times from different initial rankings, and the highest scoring network was found in 70% of cases.

To determine the hierarchy without the effects of inhibition, we excluded pairs having inhibition in either direction when calculating the optimality score and then repeated the optimization procedure described above. In this case, as there were multiple networks with the same score, we averaged the hierarchy levels across all unique rankings (Supplementary Table 1).

**Calculation of correlations between inhibition and invasion.** To test whether residents that inhibit invaders are less likely to be invaded, we constructed a $2 \times 2$ contingency table by noting for each pair of species, A and B, whether A inhibits B or not, and whether B invades A or not. We computed the ratio of invasions to non-invasions for the cases with and without inhibition. We then took the ratio of the two ratios as a measure of relative enrichment for invasion in cases with inhibition. To calculate the statistical significance of the association between inhibition and invasion, we compared the observed enrichment to that expected for random inhibition networks. During randomization of the inhibition matrix, the number of each type of pairwise link (0, 1 or 2 inhibitions) was kept constant. Similarly, to test whether an invader inhibiting a resident is more likely to invade, we constructed a $2 \times 2$ contingency table by noting for each pair of species, A and B, whether A inhibits B or not, and whether A invades B or not.

**Controlling for downward pointing tendency of inhibitions.** To determine whether inhibitions play a role in invasions independently of (or in addition to) the downward bias, we repeated the above analysis focusing only on pairs for which $3 \geq h_A - h_B > 0$, where $h_A$ and $h_B$ are the hierarchical levels of strains A and B in the invasion network. This controls for the fact that the ratio of pairs with $h_A \leq h_B$ and $h_A > h_B$ is very different for cases with and without inhibition, which might have led to spurious correlations between inhibition and invasion (that is, Simpson's paradox). Inhibitory interactions were randomly permuted only within the pairs considered, while preserving the number of inhibitions pointing up or down the hierarchy. Unfortunately, this approach could not be used to confirm with high statistical confidence that inhibition helps to resist invasion independently of the downward bias, because there are very few invasions or inhibitions against the hierarchy (Supplementary Fig. 7b). Nevertheless, invasions against the hierarchy were less frequent when the invaders were inhibited ($P = 0.07$).

**Analysis of inhibition's role in bistability.** Bistability was observed in 51% of pairs having an inhibitory interaction and in 34% of pairs without an inhibitory interaction. To test whether bistable pairings were enriched in cases with inhibition, we randomized the inhibition matrix while maintaining the number of each type of pairwise link (0, 1 or 2 inhibitions). In 2.8% of random inhibition networks, the enrichment for bistability in pairs with inhibition relative to pairs without inhibition was more than the observed value, corresponding to the $P$-value reported in the Results.

**Proving requirement of pairwise bistability for coexistence.** We will show that coexistence through the interplay between antibiotic production and degradation requires at least one bistable pair in the 'Mixed Inhibition-Zone Model' introduced in ref. 10 for the case where antibiotic producers do not derive immediate benefit from inhibiting neighbours.

Coexistence requires the fastest growing species to be inhibited by another community member. If this were not the case, it would have the highest fitness for any combination of species abundances, and therefore will unconditionally outcompete all other species. Let species 1 be the species with the highest growth rate and species 2 be the species that inhibits it most strongly. We will show that if species 1 and 2 are a part of a coexisting community, then they are in a bistable relationship.

To calculate the invasibility relationships between species 1 and 2, we set the abundances of all other species to zero and obtain the following equations for the dynamics (in accordance with the notation used in ref. 10):

$$f_1 = g_1 e^{-X_2 K_{P2}}$$

$$f_2 = g_2 e^{-X_1 K_{P1}}$$

$$X_i(t+1) = \frac{X_i(t) f_i(t)}{\sum_j X_j(t) f_j(t)},$$

where $\{X_i\}$ are the relative species abundances, $\{g_i\}$ are the growth rates, $\{f_i\}$ are the fitness values for given species abundances, $K_{P1} \geq 0$ and $K_{P2} > 0$ are the areas of inhibition caused by species 1 and 2.

To determine if species $i$ can invade species $j$, we set $X_i \to 0$ and $X_j \to 1$. The conditions for bistability (mutual non-invasion) of 1 and 2 are therefore:

$g_2 e^{-K_{P1}} < g_1$ and $g_1 e^{-K_{P2}} < g_2$.

The first condition is satisfied by construction because $g_2 < g_1$. Therefore, species 1 and 2 are not bistable if $g_1 e^{-K_{P2}} > g_2$.

The minimum fitness of species 1 over all possible abundances $\{X_i\}$ of the coexisting species is min $f_1 = g_1 e^{-K_{P2}}$ because by construction species 2 is the species that inhibits species 1 the strongest. At the same time, the maximum fitness of species 2 over all possible abundances $\{X_i\}$ of the coexisting species is max $f_2 = g_2$. Therefore, lack of bistability between species 1 and 2 implies min $f_1 >$ max $f_2$, which means that species 1 is unconditionally outcompeting species 2 in contradiction to our assumption that the species are part of a coexisting community.

Therefore, every coexisting community has at least one bistable pair. The proof does not depend on the exact functional form of antibiotic inhibition.

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

## Acknowledgements

We are grateful to John Barkei and Amelia Remiarz for assisting with experiments related to this project. We thank Sri Ram, Hannah Haight, Sailendharan Sudakaran, Michael Sullivan and Ye Xu for numerous helpful discussions. We also thank George Petry for helping to design and manufacture the device used for coating the insides of tubes with agar. We acknowledge support from the Simons Foundation, Targeted Grant in the Mathematical Modeling of Living Systems Award 342039, the National Science Foundation Grant DEB 1457518 and the National Institute of Food and Agriculture, US Department of Agriculture, Hatch project 1006261.

## Author contributions

Both authors contributed extensively to this work.

## Additional information

**Competing financial interests:** The authors declare no competing financial interests.

**How to cite this article**: Wright, E. S. & Vetsigian K. H. Inhibitory interactions promote frequent bistability among competing bacteria. *Nat. Commun.* 7:11274 doi: 10.1038/ncomms11274 (2016).

