## [Peer review file · Nature Communications]

Reviewers' Comments:

Reviewer #1 (Remarks to the Author)

Multi-species microbial communities play a vital role in the health of the planet and of the human body. Progress in understanding the functional role that these communities play relies on developing knowledge regarding how these communities assemble. In this study, the authors perform fascinating experiments to characterize the pair-wise competitive outcomes among a group of soil bacteria. They find a surprisingly large amount of bistability, in which there is survival of the common. I found the data to be of high-quality, the analysis to be solid, and the writing to be clear. This paper will be of interest to a significant community, and I believe that it could be published essentially as is. However, below I offer some suggestions for strengthening the paper.

I think that it would be nice if the authors somewhere plotted the total density at the end of each cycle (Supp Fig 2 shows fractions). In particular, do the populations reach saturation over the course of each cycle?

Supplementary Figure 2 shows sample data regarding the invader and resident fraction as a function of time. Based on the starting fraction of invader and the sensitivity of the sequencing, my understanding is that the invader frequency often starts out below the detection limit. I believe that this is ok, as the authors confirmed in each case that they were adding a given cfu of the invader, but the figure legend of Supp Fig 2 should probably indicate this fact. Also, the authors may want to mention that they can visually see the colonies of the invader in the initial runs.

What led to the 1% error in sequencing? How different are the different rpoB sequences?

Have the authors tried to determine whether growth rate or saturating density in monoculture predicts the position in hierarchy?

Minor points:

line 25: imigrants -> immigrants

line 66: Why frozen? I think that the text is fine as it is but it made me want to read the Materials to figure this out, so it might be distracting to many readers.

line 82: "mutual invasions imply coexistence..." We have seen cases in which species seem to invade each other and drive the other extinct. Assuming that in the authors data this is not what is happening then I think that the sentence can be left as it is.

line 157: My understanding of the authors' Nature paper (and other models) was that the breakdown of antibiotics facilitated coexistence even in the absence of bistability. The "consistent" statement therefore doesn't seem to follow.

Reviewer #2 (Remarks to the Author)

The authors describe a very nice test for the existence of frequency-dependent outcomes in model bacterial communities, finding that a substantial proportion of interactions between organisms can result in bistability, with the 'winner' dependent on initial abundance and type of interaction. Effects of priority are generally less considered for bacterial community assembly than for other organisms; the vast majority of studies of bacterial communities focus on the roles of environmental filtering as a way to explain community patterns and ignore other aspects of ecological theory. The authors present a case for the outcomes of community dynamics being determined largely by the initial abundance of organisms, particularly when interactions between the organisms are strong (inhibition). These types of controlled studies are extremely valuable for testing mechanistic hypotheses regarding the processes underlying community assembly as well as drawing attention to the likelihood that niche-based models of community assembly have limited explanatory power for microbial communities. The results may help explain patterns observed in recent field studies where soil bacterial and fungal communities were found to exhibit much higher levels of dissimilarity relative to predictions of models assuming that

environmental filtering and/or dispersal were the primary drivers of assembly:

- Powell JR, Karunaratne S, Campbell CD, et al. (2015) Deterministic processes vary during community assembly for ecologically dissimilar taxa. *Nat Commun* 6:8444. doi: 10.1038/ncomms9444
- Powell JR, Bennett AE (2016) Unpredictable assembly of arbuscular mycorrhizal fungal communities. *Pedobiologia*. doi: 10.1016/j.pedobi.2015.12.001

I found the experiments to be designed, executed, and analysed well. The manuscript represents a substantial contribution to the field of microbial ecology.

The manuscript is very well written and the figures are clear and very helpful for understanding the text. I only have a few very minor editorial comments.

line 42: duplication of 'rock'

line 69: I think a more precise definition of invasion than 'substantially increased in abundance' is necessary here. This definition is not easy to find in the methods.

line 240: inoculums -> inocula

Reviewer #1 (Remarks to the Author):

Multi-species microbial communities play a vital role in the health of the planet and of the human body. Progress in understanding the functional role that these communities play relies on developing knowledge regarding how these communities assemble. In this study, the authors perform fascinating experiments to characterize the pair-wise competitive outcomes among a group of soil bacteria. They find a surprisingly large amount of bistability, in which there is survival of the common. I found the data to be of high-quality, the analysis to be solid, and the writing to be clear. This paper will be of interest to a significant community, and I believe that it could be published essentially as is. However, below I offer some suggestions for strengthening the paper.

We thank the reviewer for the positive comments and very constructive suggestions.

I think that it would be nice if the authors somewhere plotted the total density at the end of each cycle (Supp Fig 2 shows fractions). In particular, do the populations reach saturation over the course of each cycle?

Relative abundances of the resident and invader strains were determined using next-generation sequencing, and therefore we did not collect absolute abundance data at each time point. However, we allowed the *Streptomyces* to grow for twelve days before harvesting their spores, which is about twice as long as required for the strains to sporulate. Therefore, we believe that the communities were grown to near the point of saturation, although we have not shown this experimentally.

Supplementary Figure 2 shows sample data regarding the invader and resident fraction as a function of time. Based on the starting fraction of invader and the sensitivity of the sequencing, my understanding is that the invader frequency often starts out below the detection limit. I believe that this is ok, as the authors confirmed in each case that they were adding a given cfu of the invader, but the figure legend of Supp Fig 2 should probably indicate this fact. Also, the authors may want to mention that they can visually see the colonies of the invader in the initial runs.

We thank the reviewer for this suggestion. We adjusted Supplemental Figure 2 accordingly.

What led to the 1% error in sequencing? How different are the different *rpoB* sequences?

The *rpoB* sequences differ by at least 4 nucleotides in the 80 nucleotide region that was sequenced (the median difference was 12 nucleotides). The multiplexing Illumina i5 and i7 indices were at least 4 nucleotides different (out of 8 per index). This information has now been added to the Methods.

Given the differences between *rpoB* sequences and between indices, nucleotide substitution errors were not a significant cause of read misassignments at the substitution error rates we estimated. The misassignment error rate we referred to is due to reading a wrong multiplexing index. Such cross-indexing errors were explained well by a simple statistical model as described in the Methods, which enabled background subtraction. Basically, an incorrect index can be read with a small probability in proportion to its overall abundance. We experimentally confirmed that the cross-indexing errors occur during sequencing (and are not for example due to primer cross-contamination) by using an independent index pair and allele combination that was amplified separately before sequencing and used independently ordered primers. Although we are unaware of published instances of dual-index cross-indexing errors occurring on the Illumina platform, this might be due to the fact that it is uncommon to include controls to detect such problems.

Have the authors tried to determine whether growth rate or saturating density in monoculture predicts the position in hierarchy?

Both growth rate and yield of each strain grown by itself were poor predictors of hierarchical position, although growth rate did display a modest correlation ($R^2=0.17$) with hierarchy level. This is shown in a new Supplemental Fig. 5, and now noted in the main text.

Minor points:

line 25: imigrants -> immigrants

This misspelling has been fixed.

line 66: Why frozen? I think that the text is fine as it is but it made me want to read the Materials to figure this out, so it might be distracting to many readers.

We have removed the reference to freezing the communities from the Results section, and elaborated on why the communities were frozen in the Methods section.

line 82: "mutual invasions imply coexistence..." We have seen cases in which species seem to invade each other and drive the other extinct. Assuming that in the authors data this is not what is happening then I think that the sentence can be left as it is.

We changed "mutual invasions imply coexistence..." to "mutual invasions are expected to lead to coexistence...". As already stated in the text, in all cases where we observed coexistence, both strains were present at the end of three growth cycles. We have not observed a case where two strains are mutually invading yet also drive the residents extinct.

line 157: My understanding of the authors' Nature paper (and other models) was that the breakdown of antibiotics facilitated coexistence even in the absence of bistability. The "consistent" statement therefore doesn't seem to follow.

Bistability was not discussed in the Nature paper. We performed a new theoretical analysis of the model (see Methods), which demonstrates that communities coexisting through this mechanism have to contain at least some pairwise bistability. The consistency statement is based on this additional analysis. We modified the text to emphasize that this is a new result: "Extending this previous work, we proved that communities coexisting through this mechanism must contain bistable pairs (see Methods)."

Reviewer #2 (Remarks to the Author):

The authors describe a very nice test for the existence of frequency-dependent outcomes in model bacterial communities, finding that a substantial proportion of interactions between organisms can result in bistability, with the 'winner' dependent on initial abundance and type of interaction. Effects of priority are generally less considered for bacterial community assembly than for other organisms; the vast majority of studies of bacterial communities focus on the roles of environmental filtering as a way to explain community patterns and ignore other aspects of ecological theory. The authors present a case for the outcomes of community dynamics being determined largely by the initial abundance of organisms, particularly when interactions between the organisms are strong (inhibition). These types of controlled studies are extremely valuable for testing mechanistic hypotheses regarding the processes underlying community assembly as well as drawing attention to the likelihood that niche-based models of community assembly have limited explanatory power for microbial communities. The results may help explain patterns observed in recent field studies where soil bacterial and fungal communities were found to exhibit much higher levels of dissimilarity relative to predictions of models assuming that environmental filtering and/or dispersal were the primary drivers of assembly:

- Powell JR, Karunaratne S, Campbell CD, et al. (2015) Deterministic processes vary during community assembly for ecologically dissimilar taxa. *Nat Commun* 6:8444. doi: 10.1038/ncomms9444
- Powell JR, Bennett AE (2016) Unpredictable assembly of arbuscular mycorrhizal fungal communities. *Pedobiologia*. doi: 10.1016/j.pedobi.2015.12.001

We are grateful to the reviewer for the supportive comments and for pointing out the connection between our results and the two recent field studies. We have highlighted this connection along with the suggested citations in a new sentence in the Discussion section.

I found the experiments to be designed, executed, and analysed well. The manuscript represents a substantial contribution to the field of microbial ecology.

The manuscript is very well written and the figures are clear and very helpful for understanding the text. I only have a few very minor editorial comments.

line 42: duplication of 'rock'

The repeated word has been removed.

line 69: I think a more precise definition of invasion than 'substantially increased in abundance' is necessary here. This definition is not easy to find in the methods.

We have clarified the definition of invasion on this line of the text.

line 240: inoculums -> inocula

This misspelling has been corrected.